# Molecular Remodeling of the Sperm Proteome Following Varicocele Sclero-Embolization: Implications for Semen Quality Improvement

**DOI:** 10.3390/proteomes13030034

**Published:** 2025-07-15

**Authors:** Domenico Milardi, Edoardo Vergani, Francesca Mancini, Fiorella Di Nicuolo, Emanuela Teveroni, Emanuele Pierpaolo Vodola, Alessandro Oliva, Giuseppe Grande, Alessandro Cina, Roberto Iezzi, Michela Cicchinelli, Federica Iavarone, Silvia Baroni, Alberto Ferlin, Andrea Urbani, Alfredo Pontecorvi

**Affiliations:** 1Complex Operative Unit of Internal Medicine, Endocrinology and Diabetology, Department of Translational Medicine and Surgery, Fondazione Policlinico Universitario A. Gemelli IRCCS, 00168 Rome, Italy; domenico.milardi@policlinicogemelli.it (D.M.); emanuelepierpaolo.vodola@guest.policlinicogemelli.it (E.P.V.); alessandro.oliva1@unicatt.it (A.O.); alfredo.pontecorvi@policlinicogemelli.it (A.P.); 2International Scientific Institute Paul VI, Università Cattolica del Sacro Cuore, Fondazione Policlinico Universitario A. Gemelli IRCCS, 00168 Rome, Italy; francesca.mancini@policlinicogemelli.it (F.M.); fiorella.dinicuolo@policlinicogemelli.it (F.D.N.); 3Department of Medicine and Translational Surgery, General Pathology Section, Università Cattolica del Sacro Cuore, 00168 Rome, Italy; 4Department of Basic Biotechnological Sciences, Intensive and Perioperative Clinics, Università Cattolica del Sacro Cuore, 00168 Rome, Italy; emanuela.teveroni@guest.policlinicogemelli.it (E.T.); michela.cicchinelli@unicatt.it (M.C.); federica.iavarone@unicatt.it (F.I.); silvia.baroni@policlinicogemelli.it (S.B.); andrea.urbani@policlinicogemelli.it (A.U.); 5Unit of Andrology and Reproductive Medicine, Department of Systems Medicine, University Hospital of Padova, 35122 Padova, Italy; grandegius@gmail.com (G.G.); alberto.ferlin@unipd.it (A.F.); 6Department of Imaging Diagnostics, Oncological Radiotherapy, and Hematology—Institute of Radiology, Fondazione Policlinico Universitario A. Gemelli IRCCS, 00168 Rome, Italy; alessandro.cina@policlinicogemelli.it (A.C.); roberto.iezzi@policlinicogemelli.it (R.I.); 7Clinical Chemistry, Biochemistry and Molecular Biology Operations (UOC), Fondazione Policlinico Universitario A. Gemelli IRCCS, 00168 Rome, Italy

**Keywords:** varicocele, sclero-embolization, sperm proteome, PRDX, TXN, stathmin, IFT20, ADAM21, selenoprotein P, ATG9A

## Abstract

Background: Varicocele is a common condition involving the dilation of veins in the scrotum, often linked to male infertility and testicular dysfunction. This study aimed to elucidate the molecular effects of successful varicocele treatment on sperm proteomes following percutaneous sclero-embolization. Methods: High-resolution tandem mass spectrometry was performed for proteomic profiling of pooled sperm lysates from five patients exhibiting improved semen parameters before and after (3 and 6 months) varicocele sclero-embolization. Data were validated by Western blot analysis. Results: Seven proteins were found exclusively in varicocele patients before surgery—such as stathmin, IFT20, selenide, and ADAM21—linked to inflammation and oxidative stress. After sclero-embolization, 55 new proteins emerged, including antioxidant enzymes like selenoprotein P and GPX3. Thioredoxin (TXN) and peroxiredoxin (PRDX3) were upregulated, indicating restoration of key antioxidant pathways. Additionally, the downregulation of some histones and the autophagy-related protein ATG9A suggests a shift toward an improved chromatin organization and a healthier cellular environment post-treatment. Conclusions: Varicocele treatment that improves sperm quality and fertility parameters leads to significant proteome modulation. These changes include reduced oxidative stress and broadly restored sperm maturation. Despite the limited patient cohort analyzed, these preliminary findings provide valuable insights into how varicocele treatment might enhance male fertility and suggest potential biomarkers for improved male infertility treatment strategies.

## 1. Introduction

Varicocele is defined as the abnormal enlargement of the testicular veins of the pampiniform plexus, typically resulting from retrograde blood flow [1]. It is recognized as one of the primary treatable causes of male infertility [2,3]. The global burden of male infertility is substantial, with varicocele being diagnosed in 35–50% of men with primary infertility and up to 80% of those with secondary infertility. Nevertheless, approximately 85% of men with varicocele remain fertile and do not encounter difficulties in conceiving [4].

Although varicocele is not universally causative of infertility, several studies have established its association with impaired sperm function and elevated oxidative stress levels [2,5,6], suggesting a strong link to subfertility. The decision to treat varicocele remains a matter of clinical debate. According to the European Association of Urology (EAU), treatment is recommended when the varicocele is palpable, semen parameters are impaired in the context of couple infertility, and the female partner has confirmed normal fertility [7]. More recent evidence supports intervention in clinically significant cases—particularly in patients with grade II–III varicocele and no other identifiable causes of male factor infertility [8].

Among treatment options, microsurgical subinguinal varicocelectomy is the most widely performed procedure due to its low recurrence rate [9]. However, percutaneous sclero-embolization has emerged as a minimally invasive and cost-effective alternative, offering shorter recovery time and fewer complications, with outcomes comparable to surgical methods [10]. Despite these therapeutic advances, the ideal timing for varicocele repair remains unresolved, as only 50–80% of men experience improvement in semen parameters post-treatment [11,12,13,14]. Therefore, there is a pressing need to elucidate the molecular mechanisms underlying sperm dysfunction in varicocele and to determine how these mechanisms respond to treatment.

The pathophysiology of varicocele-associated infertility is complex and not yet fully understood. However, recent advances in molecular biology—particularly in the fields of genetics, epigenetics, transcriptomics, and proteomics—are providing new perspectives on the condition. Proteomics, defined as the comprehensive study of the full complement of proteins within a cell at a given time, considers factors such as protein localization, interactions, post-translational modifications, and turnover. Unlike transcriptomics, which captures gene expression, proteomics reflects functional cellular activity and offers greater insight into biological complexity relevant to personalized medicine [15].

In the context of male infertility, proteomic approaches have been extensively applied to investigate the molecular basis of sperm function, hormonal regulation [16], and to identify novel biomarkers for infertility-related disorders [17]. Specifically, proteomic profiling of seminal fluid and spermatozoa in infertile men with varicocele has revealed numerous differentially abundant proteins associated with sperm structure and function. Compared to fertile men, infertile patients with varicocele show a higher number of under-expressed sperm proteins. Some proteins, such as AKAP, CABYR, SEMG1, APOPA1, ACR, RSPH1, SPA17, RSPH9, and DNAH17, are downregulated and are known to be involved in fertilization processes. Conversely, proteins such as GSTM3, DLD, NPC23, TGM4, ODF2GPR64, HIST1H2BA, PSM8, and PARK7 are upregulated in men with unilateral varicocele and are linked to poor sperm quality and function [18]. These proteins are primarily involved in mitochondrial function, oxidative stress regulation, and detoxification pathways, directly affecting sperm motility and morphology.

Comparative studies have also highlighted proteomic differences between unilateral and bilateral varicocele cases, showing that bilateral varicocele is associated with more severe alterations in sperm protein profiles and a greater impact on semen quality [19]. Furthermore, proteomic analyses of seminal fluid before and after varicocele treatment have identified differential abundance in proteins related to oxidative damage, energy metabolism, protein stabilization, spermatogenesis, and sperm motility [20,21,22].

Notably, proteins such as heat shock protein A5 (HSPA5), superoxide dismutase 1 (SOD1), and the δ-subunit of mitochondrial ATP synthase (ATP5D)—all associated with DNA integrity and sperm motility—have been reported to increase following surgical varicocelectomy [23]. To date, no studies have investigated changes in the sperm proteomic profile of patients undergoing percutaneous varicocele treatment, particularly among those who exhibit improved semen parameters (good responders). This study aims to evaluate the differential proteomic profile of spermatozoa—using mass spectrometry—in these patients before and after percutaneous embolization of the spermatic vein. The objective is to enhance our understanding of the molecular mechanisms underlying the beneficial effects of varicocele treatment and to identify potential biomarkers that could improve the clinical management of male infertility. To validate the molecular signature of successful varicocele treatment, a future study will involve a larger patient cohort and compare the findings with profiles from poor responders.

## 2. Materials and Methods

### 2.1. Patients and Study Design

Five male patients affected by varicocele who underwent left-varicocele sclero-embolization, were examined in the Endocrinology Unit at Fondazione Policlinico Universitario Agostino Gemelli, IRCCS Rome, Italy, and showed an improvement in seminal parameters (the definition of “good responder” will be given in the section “semen analysis”) were enrolled in this study. The study was conducted in accordance with the declaration of Helsinki, as revised in 2013, and all subjects provided written, informed consent. Approval for sperm collection and analysis was obtained from the Fondazione Policlinico Universitario A. Gemelli, IRCCS, local ethics committee (protocol ID 4450, date of approval: 29 September 2021). Human semen samples were obtained from the five patients diagnosed with varicocele before sclero-embolization and 3 and 6 months after the procedure. Only adult men aged 20 to 50 affected by varicocele grades III or IV on the Sarteschi scale, indicating surgical intervention (for dispermia or pelvic pain) as per the European Association of Urology (EAU) [7], were included. The exclusion criteria included fever 90 days before semen collection, the use of drugs, alcoholism, smoking, uro-genital infection, active malignancy, endocrinopathies (and their treatments), a history of cryptorchidism, or obesity (body mass index > 30 kg/m^2^). A thorough physical examination, conducted by endocrinologists, involved patients providing their medical history. Varicocele diagnosis was established through scrotal palpation in accordance with guidelines and confirmed by Doppler scrotal ultrasound. Patients who met the inclusion criteria provided a semen sample through masturbation after 3–5 days of ejaculatory abstinence. Patients subsequently underwent percutaneous radiologically guided sclero-embolization with 2 cc of 3% Atossisclerol, 2 cc of 95% ethyl alcohol, and a metal alloy coil (IDC, MRI-compatible).

Follow-up assessments were performed at 3 and 6 months post-procedure to monitor improvements in semen quality and individual patient characteristics. At the first follow-up, the procedure’s success was evaluated, with no reported cases of recurrent or persistent varicoceles. During the follow-up visit, patients provided semen samples.

### 2.2. Semen Analysis

Semen samples were obtained through masturbation following a 3–5-day period of sexual abstinence and were assessed following the World Health Organization (WHO) guidelines from 2021. The evaluation included various features, including pH, volume (ml), sperm count (10^6^/mL), total sperm count (10^6^/ejaculate), total and progressive motility (%), morphology (percentages of normal and abnormal forms), and leukocyte count (10^6^/mL). The seminology laboratory actively participated in both internal and external quality control tests. In accordance with the WHO 2021, semen samples need liquefaction at 37 °C for 30 min before analysis. Sperm cells were isolated from other semen components by layering liquefied semen on a 50% Percoll^TM^ solution (GE Healthcare, Uppsala, Sweden) density gradient equilibrated with HEPES-buffered Ham’s F10 medium (Gibco, Life Technologies, Paisley, UK) supplemented with sodium bicarbonate (0.2% NaHCO_3_; *w*/*v*; Merck, Darmstadt, Germany). The samples were then centrifuged at room temperature for 30 min at 400× *g*. The resulting pellet of purified sperm was collected, washed in PBS, and centrifuged for 15 min at 400× *g*. Subsequently, the sperm cells were lysed in 2% SDS and immediately stored at −80 °C for proteomic and Western blot experiments. As a composite variable, total motile sperm count (TMSC) was calculated as follows: progressive motility × total sperm count. TMSC was adopted as a parameter of good response to the treatment. All the five patients included were considered good responders, as their TMSC increased more than 50% from basal evaluation after 3 months and/or 6 months.

### 2.3. Proteomic Analysis

For the global proteomic profiling study, pooled sperm lysate samples (10 µg per patient) were prepared from the varicocele group at three time points: before treatment (N = 5), and at 3 months (N = 5) and 6 months (N = 5) after varicocelectomy. Each pooled sample was analyzed in triplicate. Protein lysates obtained from purified sperm were subjected to the FASP protocol [24,25] via Microcon 10 k centrifugal ultrafiltration units (Merck, Darmstadt) operating at 10,000× *g*. Aliquots containing 50 μg (in 50 μL 2% SDS) of total pooled protein were combined with 100 μL of 8 M urea in 0.1 M Tris/HCl, pH 8.5 (UA), within the ultrafiltration unit. After centrifugation at 20 °C for 15 min at 10,000× *g*, the eluates were discarded. Subsequently, 200 μL of UA was pipetted into the filtration unit, and the units underwent centrifugation (repeated three times). Next, 100 μL of 8 mM DTT was added, and the mixture was incubated at 56 °C for 15 min. After centrifugation for 10 min at 10,000× *g*, 100 μL of UA was added, and the mixture was centrifuged for 30 min at 13,000× *g*, with the eluates discarded at each step.

In the next steps, 100 μL of 50 mM IAA in UA was introduced to the filters, and the samples were incubated in darkness for 20 min at room temperature. The samples were then centrifuged for 10 min at 13,000× *g*. The filters were washed with 100 μL of UA and centrifuged for 30 min at 11,000× g. Excess IAA was quenched by adding 100 s of DTT, followed by incubation for 15 min at 56 °C. After centrifugation for 10 min at 13,000× *g*, 100 μL of UA was added to the filters to remove DTT, followed by centrifugation for 30 min at 13,000× *g*. Subsequently, 100 μL of 50 mM NH_4_HCO_3_ was added, and the mixture was centrifuged for 10 min at 11,000× *g* (repeated two times).

New collection tubes were used, and 50 μL of trypsin (MS grade) at an enzyme-to-protein ratio of 1:50 *w*/*w* was added. The mixture was incubated at 37 °C for 18 h in a wet chamber. The released peptides were collected by centrifugation at 11,000× *g* for 30 min, followed by two washes with 50 μL of 50 mM NH_4_HCO_3_. The mixture was then blocked with 20 μL of 1% TFA.

LC-MS analyses of samples were performed in triplicate using the UltiMate 3000 RSLCnano System coupled to Orbitrap Fusion Lumos Triibrid MS detector with EASY-Spray nanoESI source (Thermo Fisher Scientific, Waltham, MA, USA).

The ion source type, NSI, worked in positive polarity (voltage 1800 V); ion transfer tube temp. was 275 °C. The following MS parameters were set: the acquisition of high-resolution MS/MS spectra was carried out in data-dependent scan mode (DDS) with Orbitrap as detector, a resolution of 120,000 in a 375–1500 m/z range of acquisition and higher-energy collisional dissociation (HCD) fragmentation.

For bottom-up analyses, the chromatographic column used was an EASY-Spray column (15 cm × 50 μm ID, PepMap C18, 2 μm particles, 100 A pore size) coupled with an Acclaim PepMap100 cartridge (C18, 5 μm, 100 A, 300 μm i.d. × 5 mm) from Thermo Fisher Scientific. Gradient elution was performed using eluent A (0.1% FA, *v*/*v*) and solvent B (ACN:FA 99.9:0.1, *v*/*v*) in the following steps: 5% B (2 min), from 5 to 55% B (130 min), from 55 to 99% B (15 min), 99% B (10 min), from 99 to 5% B (2 min), and (vi) 5% B (13 min) at a flow rate of 0.3 μL/min. A 5 μL injection, corresponding to 0.5 μg of total protein after dilution with 0.1% (*v*/*v*) FA aqueous solution, was used. Chromatographic separations were performed in triplicate at 40 °C.

LC-MS and MS/MS data were elaborated by Proteome Discoverer software (version 2.4, Thermo Fisher Scientific), based on SEQUEST HT cluster as search engine against the Swiss-Prot Homo Sapiens proteome (UniProtKb, Swiss-Prot, homo + sapiens). The identifications were validated by the Percolator node, with the strict target value of false discovery rate (FDR) set at 0.01 and the relaxed value at 0.05. Identification data were filtered for high confidence and peptide rank 1.

To handle missing values, we considered the mean abundance of each protein across the technical triplicates. Only proteins that were quantified in at least two out of the three technical replicates were included in the analysis. This approach was used to ensure data reliability while minimizing the impact of missing values.

Normalization across samples was performed using the area of unique peptide peaks, which were normalized to account for variability between samples. This strategy allows for more accurate quantification by focusing on peptides uniquely assigned to each protein.

### 2.4. Western Blot

For Western blot analysis performed to validate the proteomic data, the sperm lysates were sonicated for 15 s at 50% amplitude on ice and centrifuged for 15 min at 15,000× *g* at 4 °C. Aliquots of the samples were denatured in Laemmli buffer followed by heating at 95 °C for 5 min. Equal amounts (10 µg) of protein were loaded and separated by SDS-PAGE, using 4–12% bis-tris gels (mPAGE^TM^ Millipore, Burlington, MA, USA) in MOPS SDS running buffer (Millipore, Burlington, MA, USA) according to the manufacturer’s instruction. Electro-transfer of proteins was performed at 100 Volts for 90 min. Transfer efficiency of proteins transferred onto PVDF membranes (IPVH00010, Immobilon Millipore, Burlington, MA, USA) was assessed by Red Ponceau staining (0.1% Ponceau in 5% acetic acid, Sigma-Aldrich, St. Louis, MO, USA). Membrane blocking was performed for 1 h with blocking reagent according to the manufacturer’s instruction (SuperBlock^TM^ T20 PBS, Thermofisher, Rockford, IL, USA). Membranes were then incubated overnight at 4 °C with the following primary antibodies: rabbit polyclonal α-PRX1, PRX3, PRX6 and TXN (Genetex, Alton Pkwy Irvine, CA, USA), rabbit monoclonal α-H2B (Millipore Burlington, MA, USA), and mouse monoclonal α-Tubulin (Sigma-Aldrich, St. Louis, MO, USA). Membranes were washed 15 min in PBS Tween (0.5%) and incubated in specific horseradish peroxidase-conjugated HRP-IGG diluted 1:5000 (Biorad, Hercules, CA, USA) in blocking buffer. Membranes were washed 15 min in PBS Tween (0.5%) and 15 min in PBS. The membranes were finally developed via enhanced chemiluminescence (ECL Westar, Cynagen, Bologna, Italy), and the signals were detected via a chemiluminescence imaging system, Alliance 2.7 (UVITEC, Cambridge, UK), and quantified via Alliance V_1607 software. The levels of target proteins were estimates versus tubulin level as loading control.

### 2.5. Statistical and Bioinformatics Analysis

The data are presented as the mean ± standard deviation (SD). Statistical analysis employed either a paired two-tailed Student’s *t*-test, one-way ANOVA (for seminal parameters) or a one-sample *t*-test; the latter was applied when the mean value of the control group was arbitrarily set to one. Significance was considered at *p* < 0.05. The statistical software used for these analyses was GraphPad Prism 10.3.1.

Bioinformatics analysis was performed through differential abundance analysis and pathway enrichment. Protein abundances were normalized through quantile normalization (MATLAB Statistics and Machine Learning Toolbox, R2022b version), and fold change (FC) values were calculated for the Post3M/Pre and Post6M/Pre ratios. Proteins that met the criteria of a LogFC > 1.5 were considered significantly upregulated, whereas those with a LogFC < −1.5 were considered significantly downregulated. In differentially abundant protein (DAP) analysis, *p*-values were corrected for multiple testing using the Benjamini–Hochberg procedure to control the false discovery rate (FDR). The significance threshold of log2FC > 1.5 was chosen empirically, based on commonly used cut-offs in the literature for identifying biologically meaningful changes in protein expression.

Pathway enrichment was performed via the Reactome database (http://reactome.org, accessed on 31 May 2024) by overrepresentation analysis based on FDR. To construct a protein—protein interaction network for the identified sperm proteins, we employed the Search Tool for the Retrieval of Interacting Genes/Proteins (STRING 12.0) software. The resulting network visually represents proteins as nodes, with edges (colored lines) connecting them to depict their interactions.

For the STRING analysis, we used a minimum required interaction score of 0.7 (high confidence) to construct the protein–protein interaction (PPI) network. The network was further filtered to include only experimentally validated and database-annotated interactions. For the Reactome pathway enrichment analysis, default settings were used, with a significance threshold of FDR < 0.05.

## 3. Results

### 3.1. Patient Characteristics and Semen Analysis

The characteristics of the five patients included in the study are shown in Table 1.

According to the Sarteschi Ultrasound Classification, four patients presented with left grade III varicocele, and one presented with left grade IV varicocele. The mean ± standard deviation age was 35.6 ± 8.17 years.

Seminal parameters were evaluated before (Pre), 3 (3M) and 6 months (6M) after the percutaneous procedure. In the five selected patients, varicocele sclero-embolization led to a significant increase in sperm count, morphology and total and progressive motility (TMSC). Semen volume increased, although not significantly at 3 months, reaching significance at 6 months when compared to the ones detected before the procedure (Figure 1a).

A significant increase in the sperm count and morphology was observed at both 3 and 6 months after the procedure (Figure 1b,c). Importantly, total motile sperm count (TMSC), used as a proxy indicator of a positive seminal response to treatment, showed a significant increase at both 3 and 6 months post-treatment (Figure 1d). All patients showed less than 100,000 leucocytes in semen samples. Furthermore, Figure 1e,f showed the increase in sperm number/ejaculate and TMSC in each patient pre (Spz Pre) and after 3 (Spz 3M) and 6 (Spz 6M) months from the percutaneous procedure (PzA-E).

### 3.2. Proteome Analysis

For the global proteomic profiling study, each pooled spermatozoa sample from the varicocele group before, hereafter Pre (N = 5), and 3 (N = 5) and 6 (N = 5) months after varicocele sclero-embolization (hereafter POST3M and POST6M, three and six months, respectively) was analyzed. Proteins identified under each condition, with the IDs and the relative abundance, are listed in Appendix A. In particular, we found 2204 proteins in Pre, 2240 proteins in POST3M, and 2234 proteins in POST6M samples. The heatmap showed the proteomic analysis data, illustrating protein levels across treatment groups (protein levels for each group of treatments are presented in analytical triplicate). Relative protein abundance was indicated by color, with red signifying higher and green lower abundance (Figure 2a). The heatmap illustrated marked differences in protein abundance between sperm cells collected before and at 3 and 6 months after varicocele sclero-embolization. These distinct patterns of protein abundance effectively differentiate the pre-treatment group from the post-treatment groups, suggesting that varicocele sclero-embolization induces widespread changes in protein expression over time. Figure 2b shows Venn diagrams of proteins identified in sperm cells Pre and after varicocele sclero-embolization (POST3M and POST6M). Venn proteins were listed in Appendix A. As expected, the majority of proteins were shared across all groups. As shown in Figure 2b, seven proteins were exclusively detected in the pre-treatment group, whereas 70 proteins were uniquely present in the post-treatment groups, suggesting a substantial shift in the sperm proteome following embolization. This was further quantified in the DAP (differentially abundant proteins) histogram between Pre, POST3M and POST6M (Figure 2c). In the POST3M group, 163 proteins were upregulated and 91 downregulated, while the POST6M group exhibited 174 upregulated and 94 downregulated proteins. The list of DAPs after varicocele sclero-embolization are shown in Appendix A. Importantly, 115 upregulated and 33 downregulated proteins were common to both post-treatment time points, indicating a shared core response to varicocele treatment alongside time-specific proteomic adaptations. Collectively, these findings demonstrate that varicocele sclero-embolization induces widespread and time-dependent changes in the sperm proteome, reflecting both the immediate and sustained molecular responses associated with improved semen parameters.

In the Pre group, seven proteins were uniquely identified (Figure 2b). Among that list, we highlighted stathmin, intraflagellar transport protein 20 (IFT20), selenide, and the disintegrin and metalloproteinase domain-containing protein 21 (ADAM21) (Appendix A).

Fifty-five proteins common to the POST3M and POST6M groups were uniquely present in these groups with respect to Pre samples (Figure 2b). These proteins included RNA-binding protein RO60, beta-2-glycoprotein 1 (APOH), selenoprotein P, glutathione peroxidase 3 (GPX3), testican-1, uteroglobin, plasminogen heavy chain A, and leucine-rich repeat-containing protein 59 (LRRC59). The network generated from these protein groups via STRING revealed that the major node of interaction was connected by beta-2-glycoprotein 1 (ApoH in Figure 3a).

ApoH is linked to proteins of the glutathione peroxidase family, such as selenoprotein P and GPX3 (Figure 3a). The 10 most relevant pathways, sorted by *p* value, of 55 post-VAR common proteins are shown in Figure 3b; the most representative pathways are linked to platelet activation, signaling, aggregation and degranulation (in Figure 3b, “response to elevated platelet calcium” and “platelet degranulation”). Cellular component enrichment analysis (gene ontology) revealed a significant overrepresentation of genes associated with the extracellular region, extracellular space, and exosome (Figure 3c).

The list of DAPs of 3 and 6 months after varicocele sclero-embolization are shown in Appendix A, respectively. All proteins that were differentially abundant under both conditions are listed in Appendix A. Most of the common DAP proteins (N = 115) were upregulated. The in silico-generated protein—protein interaction network generated via STRING from these results is shown in Figure 4a.

The study observed a significant increase in proteins involved in antioxidant defense, specifically thioredoxin (TXN) and thioredoxin-dependent peroxide reductase 3 (PRDX3) after surgery (Appendix A). Moreover, STRING analysis highlighted that many of the upregulated proteins were connected with SERPINA5, CRISP3, KLK proteins and EPHX2. Reactome clustering pathway analysis (Figure 4b) of DAP-upregulated proteins was performed. The most representative enriched pathways were those linked to “neutrophil degranulation, platelet degranulation, and platelet degranulation in the major pathway of response to elevated platelet cytosolic Ca^2+^”. According to the classification for biological process (gene ontology) (Figure 4c), the main upregulated proteins are predominantly represented by negative regulation of endopeptidase activity, negative regulation of hydrolase activity, and negative regulation of proteolysis (CST4, TIMP2, CSTB, SERPINF2, SERPINA5, SPINT1, WFDC2, RPS6KA3, SERPINB1, CST3, and SERPINA1). These proteins are involved in vesicular trafficking and endopeptidase activities.

Thirty-three common DAP-downregulated proteins were detected. Protein—protein interaction network analyses via the STRING database with functional enrichment identified two main clusters (Figure 5a).

The protein cluster with the highest representation is composed of different heterogeneous nuclear ribonucleoproteins. The downregulated proteins are components of the spliceosome, the molecular machine essential for RNA editing, which is critical for protein synthesis. Of interest, ATG9, a key component in autophagy, becomes significantly downregulated (Figure 5a). Furthermore, histone H3.1 abundance was significantly decreased after 6 months from sclero-embolization (Appendix A). Interestingly, H2B decrease in sperm upon surgery both at 3 and 6 months (Appendix A), although not significantly. Reactome analysis revealed that most DAP-downregulated pathways are linked predominantly to processes associated with mRNA metabolism, such as the processing of capped intron-containing pre-mRNAs, mRNA splicing, and extracellular matrix organization and the degradation of the extracellular matrix.

### 3.3. Validation of the Proteomics Data

To validate the proteomic data, Western blot analyses were performed. Figure 6a shows the ratio (FC-fold change) of POST/PRE proteomic quantifications from the varicocele group at 3 (3 M) and 6 months (6 M) after surgery for different proteins involved in antioxidant defense, such as TXN, PRDX1, PRDX3 and PRDX6.

Figure 6b shows a representative Western blot analyses for these proteins from the varicocele group before (PRE N = 4) and 3 (3M N = 3) and 6 (6M N = 4) months after varicocelectomy. Densitometric analysis (Figure 6c–f) confirmed an overall increase in these proteins that was significant for TXN. Figure 6g shows the ratio (FC-fold change) of POST/PRE proteomic quantification at 3 (3M) and 6 months (6M) for different histone proteins: H2B, H3.1, H1T and H4. Figure 6h shows the Western blot analyses for H2B protein from the varicocele group before (PRE N = 4) and 3 (3 M N = 3) and 6 M N = 4) months after varicocelectomy. Densitometric analysis (Figure 6i) confirmed a significant decrease in this protein six months after surgery.

## 4. Discussion

Varicocele is associated with disruptions in key testicular proteins, contributing to inflammation and impaired spermatogenesis [26]. Furthermore, increased oxidative stress has been described [27]. All these factors can alter spermatogenesis and impair fertility.

Varicocele treatment has been associated with improved semen quality in men experiencing infertility [28]. It has also been shown to reduce sperm DNA fragmentation, which may enhance overall sperm quality and fertility potential [29]. Additionally, treatment appears to modulate the abundance of proteins involved in key cellular pathways, potentially contributing to improved sperm function [30]. Despite these benefits, further research is necessary to fully understand the molecular mechanisms affected by varicocele and its treatment. The proteomic approach is widely employed to identify biomarkers of treatment efficacy, with a stronger focus on seminal fluid compared to sperm cells [20,21]. Agarwal and colleagues [18] investigated the spermatozoa proteomic profiles of infertile men with unilateral varicocele compared to fertile men, identifying protein alterations associated with varicocele, primarily affecting sperm motility, energy metabolism, and antioxidant defense. Moreover, Finelli and colleagues, integrating in silico proteomic analysis with in vitro validation, investigated the molecular mechanisms underlying varicocele-associated infertility. They identified mitochondrial aconitate hydratase (ACO2) and fatty acid synthase (FASN) as potential key proteins involved in DNA repair, whose dysregulated expression may contribute to the DNA damage observed in the pathophysiology of varicocele [31].

Our research focused on the longitudinal assessment of proteomic changes within the spermatozoa of patients who underwent percutaneous varicocele sclero-embolization and experienced improvements in their seminal parameters at 3 and 6 months post-procedure. In the five selected patients, varicocele sclero-embolization resulted in a significant improvement in key semen parameters, including sperm count, morphology, and motility. Notably, sperm count and morphology improved significantly at both 3 and 6 months following treatment. Importantly, the total motile sperm count (TMSC)—a recognized proxy for positive seminal response to treatment—demonstrated a significant increase at both post-treatment time points, further supporting the effectiveness of varicocele sclero-embolization in enhancing male fertility potential.

In our proteomic analysis, we pooled semen samples within each group to minimize inter-individual variability and enhance statistical power, particularly due to the limited sample size and high variability among individuals. In our study, pre-intervention patients presented seven proteins exclusively found in sperm, which were completely lost after treatment. We specifically highlighted stathmin, intraflagellar transport protein 20 (IFT20), selenide, and the disintegrin and metalloproteinase domain-containing protein 21 (ADAM21) from this list (see Appendix A). The majority of these identified proteins are associated with inflammation and are involved in intracellular trafficking, signaling, and stress response, notably impacting microtubule regulation, as observed with stathmin and IFT20 [32,33,34,35]. Furthermore, the presence of ADAM proteins, which participate in processes such as testicular microenvironment remodeling, inflammation, and oxidative stress [36], suggests their potential as markers of testicular damage in varicocele patients. Finally, the presence of selenide supports the hypothesis of a compensatory mechanism against the elevated reactive oxygen species (ROS) associated with varicocele. Indeed, selenide is involved in reducing oxidative stress and inflammation through its antioxidant and anti-apoptotic roles in germ cells and its cell-permeable form (H_2_Se) facilitates efficient intracellular ROS reduction [37,38].

Following varicocele treatment, we found 55 proteins uniquely identified in both the 3-months post-treatment and 6-months post-treatment groups compared to the pre-treatment samples. Network analysis using STRING showed that beta-2-glycoprotein 1 (ApoH) was a major interaction node, connected to proteins of the glutathione peroxidase family, selenoprotein P and GPX3. The presence of selenoproteins such as selenoprotein P and GPX3 is essential for sperm reproductive function. Selenoprotein P is important for selenium transport to the testes and for the development of functional sperm, while GPX3 contributes to antioxidant protection during sperm maturation. Disruption in their expression or function can lead to abnormal sperm structure and reduced fertility [39]. These proteins are known to be important players in the antioxidant response, even though the enriched pathway discovered after sclero-embolization refers to platelet activation. GPX3 deficiency indeed facilitates platelet aggregation, likely via disinhibition of thromboxane biosynthesis [40].

Among the DAPs, we detected a significant upregulation of others proteins involved in antioxidant defense, such as thioredoxin (TXN) and thioredoxin-dependent peroxide reductase 3 (PRDX3). These proteins’ abundance, together with PRDX1 and PRDX6, were validated by Western blot, confirming an overall increase in this pathway. TXN, along with other antioxidant enzymes such as superoxide dismutase (SOD), glutathione peroxidases (GPXs), and peroxiredoxins (PRDXs), is essential for the production of healthy spermatozoa. These antioxidant enzymes are crucial for healthy sperm production. These enzymes help maintain sperm quality by ensuring motility, capacitation, and DNA integrity, which are crucial for successful fertilization [41]. Furthermore, STRING analysis of this group of proteins revealed that several upregulated proteins were interconnected with SERPINA5, CRISP3, KLKs, and EPHX2. SERPINA5, found in sperm and reproductive tissues, is implicated in sperm function, fertilization, and male fertility [42]. CRISP3 (cysteine-rich secretory protein 3) is expressed in both seminal plasma and spermatozoa across several species. Importantly, higher levels of CRISP3, present in seminal plasma and spermatozoa, correlate positively with semen quality parameters and fertility [43]. Kallikrein-related peptidases (KLKs), abundant in seminal plasma, are crucial for semen liquefaction, sperm motility, and male fertility [44]. Moreover, epoxide hydrolase 2 (EPHX2) is involved in lipid signaling molecule metabolism and antioxidant defense, with its genetic variants and effects being studied in relation to male reproductive health and sperm function [45]. Reactome pathway analysis of the upregulated differentially abundant proteins (DAPs) highlighted enrichment in pathways related to “neutrophil degranulation,” “platelet degranulation,” and “platelet degranulation in the major pathway of response to elevated platelet cytosolic Ca^2+^”. The predominantly upregulated proteins are involved in the negative regulation of endopeptidase activity, hydrolase activity, and proteolysis (including CST4, TIMP2, CSTB, SERPINF2, SERPINA5, SPINT1, WFDC2, RPS6KA3, SERPINB1, CST3, and SERPINA1). These proteins are involved in vesicular trafficking and endopeptidase activities, which are crucial for sperm maturation, the acrosome reaction, and sperm–egg interactions. Endopeptidases play crucial roles in sperm maturation, the acrosome reaction, and sperm–egg interactions. These proteins are often involved in vesicular trafficking and the regulation of proteolytic activity, which are essential for successful fertilization. During spermatogenesis and epididymal maturation, endopeptidases and their inhibitors are involved in the processing, modification, and compartmentalization of acrosomal proteins. These modifications are essential for the sperm to acquire fertilization competence and for the correct localization of proteins needed during fertilization [46]. Precise control of endopeptidases, enzymes essential for fertilization, is crucial to prevent premature acrosomal reactions and potential sperm damage or inflammation. Protein inhibitors modulate proteolytic activity, ensuring proper timing and location [47]. Both acrosomal granules (containing enzymes for zona pellucida penetration) and sperm secretory vesicles (containing proteins for capacitation and oocyte fusion) release their contents through highly regulated membrane fusion. These proteins, particularly those within acrosomal granules, are critical for successful fertilization [48]. Varicocele is known to affect acrosomal enzyme activity. This regulation is influenced by factors such as oxidative stress, metal ion imbalance, and specifically, the activity of acrosin, a key enzyme for the acrosome reaction [49].

Among the differentially abundant proteins (DAPs), a significant downregulation of proteins associated with the spliceosome, histones, and autophagy regulation was observed, collectively suggesting a compromised spermatogenesis process. These characteristics point towards the presence of immature and poorly compacted sperm, which can subsequently lead to increased sperm DNA fragmentation, a known factor associated with poorer fertility outcomes [50]. The most highly represented protein cluster consists of components of the spliceosome, and it has been reported that its deregulation impairs spermatogenesis and leads to infertility [51]. Moreover, we found a downregulation of various heterogeneous nuclear ribonucleoproteins. The regulation of ribonucleoprotein (RNP) proteins is a key aspect of sperm maturation, influencing mRNA storage, translation, and the structural changes necessary for functional sperm development. During spermatogenesis, RNP particles store and regulate the translation of protamine mRNAs, which are essential for replacing histones and achieving the highly compacted chromatin structure of mature sperm. Disruption in RNP-mediated regulation of protamine expression is linked to abnormal chromatin packaging and impaired spermatogenesis [52].

Importantly, among the downregulated proteins upon varicocelectomy, we found proteins involved in chromatin organization, such as histones H2B, H3.1, H1T and H4. Spermatozoa exhibit unique chromatin compaction, characterized by the replacement of most histones with protamines, leading to a highly compacted DNA structure [53]. It has been reported that modifications and structural differences in H3.1 and related histones influence chromatin organization, gene expression, and sperm motility, affecting male fertility [54]. Importantly, H3.1 was involved in nucleosome stability during spermatogenesis [55]. Interestingly, we found that histone H2B decreases in sperm upon surgery. The replacement of canonical histone H2B with the testis-specific variant TH2B leads to chromatin relaxation by promoting the eviction of canonical histones from sperm. This transition to TH2B modulates nucleosome stability, facilitating chromatin-to-nucleoprotamine transition [56]. Varicocele can disrupt the normal histone-to-protamine transition during sperm packaging [57]. This imbalance is associated with infertility, abnormal sperm morphology, and reduced sperm count and motility. While a small percentage of human sperm (around 15%) naturally retain some histones post-development [58], disruptions to this packaging process, such as residual histones, can contribute to fertility problems. Specifically, varicocele-induced alterations in the histone–protamine balance are linked to adverse semen parameters and infertility. Thus, the observed reduction in histones could correlate to an improvement in the differentiation process of spermatozoa upon varicocele sclero-embolization.

Finally, our post varicocele sclero-embolization proteomic data revealed a downregulation of autophagy-related proteins, including ATG9A, a crucial protein for autophagosomal membrane expansion. Autophagy is recognized as essential for healthy sperm development and male fertility by facilitating the removal of surplus cellular components, supporting sperm maturation, and protecting germ cells [59]. In the context of varicocele, autophagy is initially activated as a protective mechanism against stress; however, sustained stress can result in dysfunctional autophagy, leading to damage in testicular cells and impaired sperm production [60]. The observed downregulation of autophagy-related protein suggests a potential restoration towards a more balanced cellular environment following the treatment.

Using a bottom-up proteomics workflow, this study quantifies relative protein abundance via tryptic peptide analysis. Nevertheless, it does not elucidate proteoforms and their modifications, given their significant role in biological regulation. To achieve a more comprehensive understanding of proteomic changes associated with successful varicocele treatment, future work will integrate top-down proteomics and targeted analysis of the differentially abundant proteins (DAPs) identified in the present study.

## 5. Conclusions

This proteomic analysis of spermatozoa from patients who responded positively to percutaneous varicocele sclero-embolization reveals that the treatment induces significant molecular changes associated with improved sperm quality and fertility potential. Pre-treatment, the protein profile reflected inflammation, oxidative stress, and disruptions in intracellular trafficking and microtubule regulation, with proteins like ADAM21 and selenide suggesting testicular damage and oxidative stress. Post-treatment, the emergence of antioxidant enzymes (selenoprotein P, GPX3) and the interconnected network centered on beta-2-glycoprotein 1 (ApoH) indicate a modulation of oxidative stress and inflammation. Furthermore, the upregulation of proteins involved in antioxidant defense (TXN, PRDX3), and fertilization (SERPINA5, CRISP3), alongside the apparent restoration of chromatin organization (through the regulation of ribonucleoprotein and histones), and autophagy regulation (downregulation of ATG9A), collectively suggest a more conducive environment for healthy sperm development and function. These findings, although preliminary, offer valuable insights into the molecular mechanisms underlying the beneficial effects of varicocele treatment and point towards potential biomarkers. Confirming these results requires future studies to validate the molecular signature of successful varicocele treatment in larger, unpooled patient cohorts. Importantly, these studies must also compare these profiles with those from poor responders.

## Figures and Tables

**Figure 1 proteomes-13-00034-f001:**
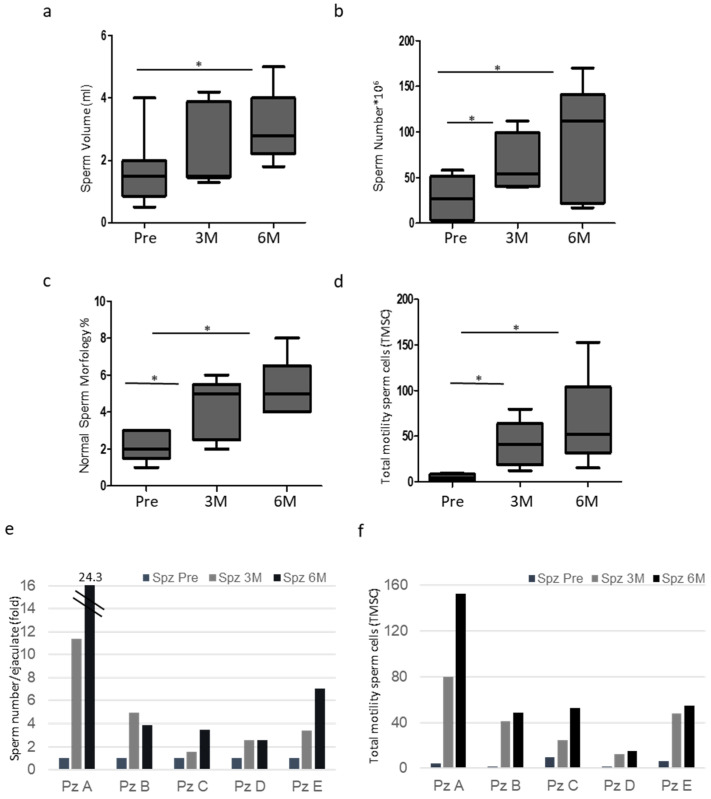
(**a**) Sperm volume of ejaculates from patients Pre, 3 (3M) and 6 months (6M) after surgery. Box plots are shown (N = 5, * *p* < 0.05, one-way ANOVA). (**b**) Total sperm number of ejaculates from patients Pre, 3 (3M) and 6 months (6M) after surgery. Box plots are shown (N = 5, * *p* < 0.05, one-way ANOVA). (**c**) Normal sperm morphology (%) of ejaculates from patients Pre, 3 (3M) and 6 months (6M) after surgery. Box plots are shown (N = 5, * *p* < 0.05, one-way ANOVA). (**d**) Total motility sperm cells (TMSC) of ejaculates from patients Pre, 3 (3M) and 6 months (6M) after surgery. Box plots are shown (N = 5, * *p* < 0.05, one-way ANOVA). (**e**) Total sperm number of matched samples in ejaculates from patients Pre, 3 (Spz 3M) and 6 months (Spz 6M) after surgery. The value of each patient’s pre-sperm sample is arbitrarily set to 1. (**f**) Total motility sperm cells (TMSC) of matched samples in ejaculates from patients Pre, 3 (Spz 3M) and 6 months (Spz 6M) after surgery.

**Figure 2 proteomes-13-00034-f002:**
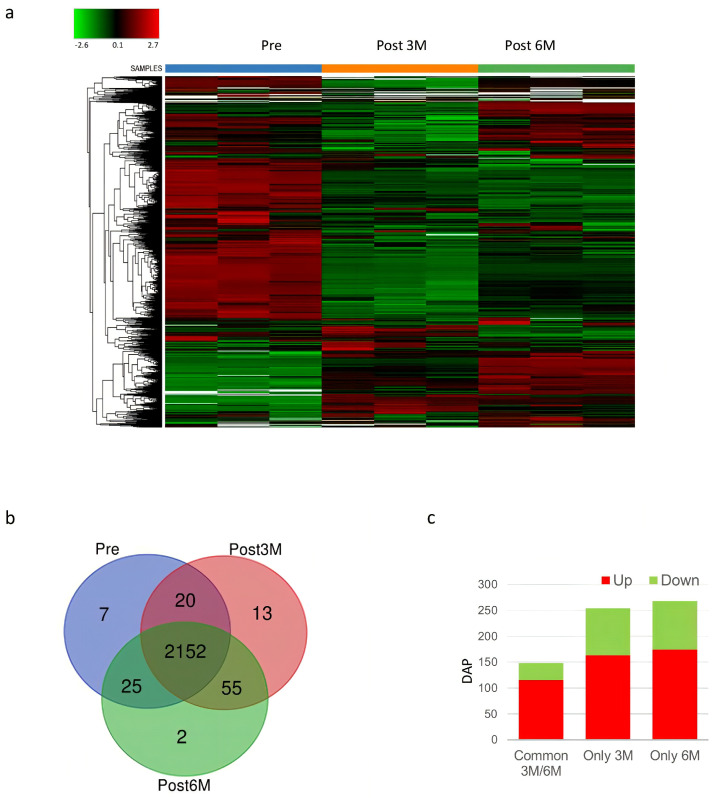
(**a**) Heatmap showing the results of the proteomic analysis; protein levels for each group of treatments are presented in analytical triplicate. The colors indicate the relative abundance in each group, with red and green indicating higher and lower abundance, respectively. (**b**) Venn diagrams resulting from grouping analysis of the proteins identified in Pre versus Post3M and Post6M groups. (**c**) Histogram showing differentially abundant proteins (DAP) between Pre and POST3M and POST6M groups. Common Post3M/6M represents the number of proteins differentially abundant in both groups while Only3M and Only6M represent the number of proteins differentially abundant in each group.

**Figure 3 proteomes-13-00034-f003:**
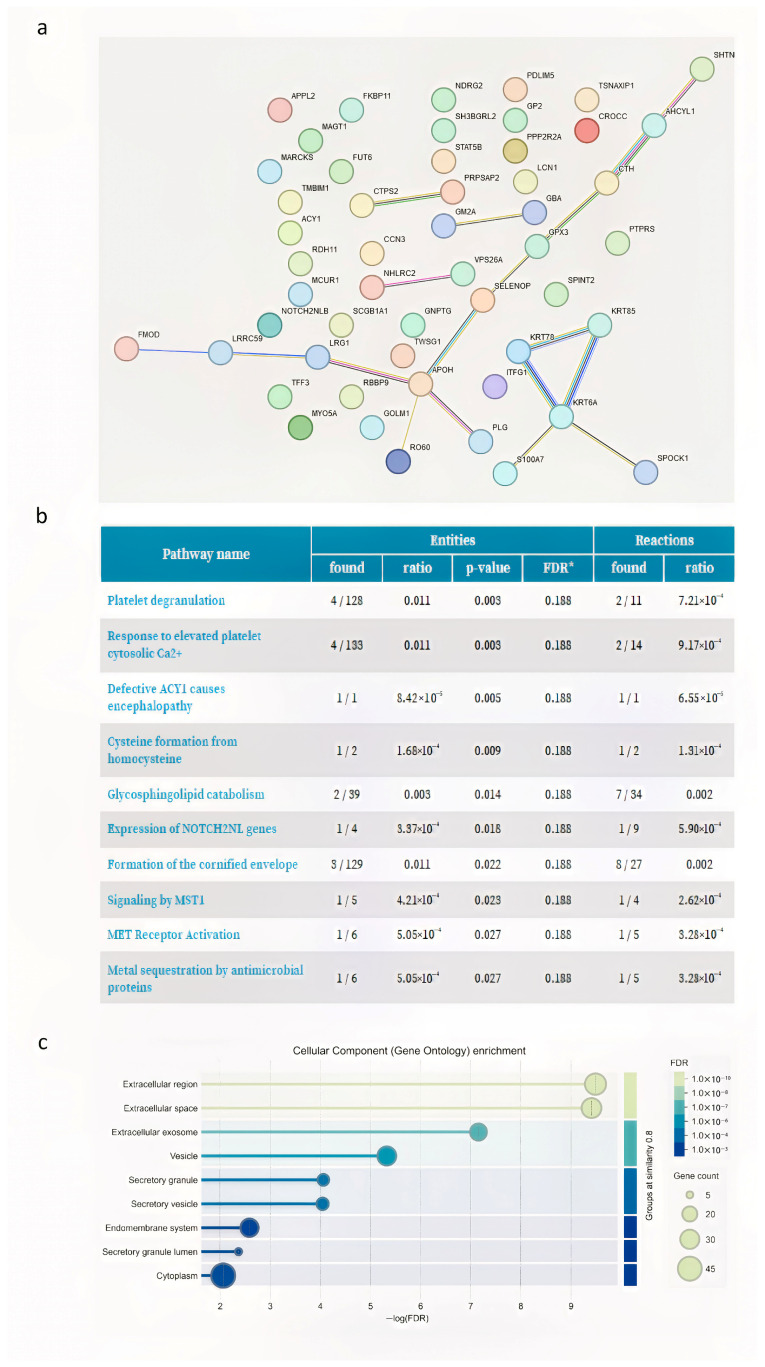
(**a**) STRING analysis shows interaction network of the 55 proteins exclusively abundant in POST3M and POST6M groups. Each node represents a protein, and each edge represents an interaction. Different colored lines represent seven types of evidence used to predict associations. Red line: fusion evidence; green line: neighborhood evidence; blue line: co-occurrence evidence; purple line: experimental evidence; yellow line: text mining evidence; light blue line: database evidence; and black line: co-expression evidence. (**b**) Table showing the 10 most relevant pathways, sorted by *p*-value, of the 55 proteins exclusively abundant in POST3M and POST6M groups (pathway enrichment was developed using Reactome database). * False Discovery Rate. (**c**) Graph showing the most relevant cellular component enrichment analysis (gene ontology) of the 55 proteins exclusively abundant in POST3M and POST6M groups.

**Figure 4 proteomes-13-00034-f004:**
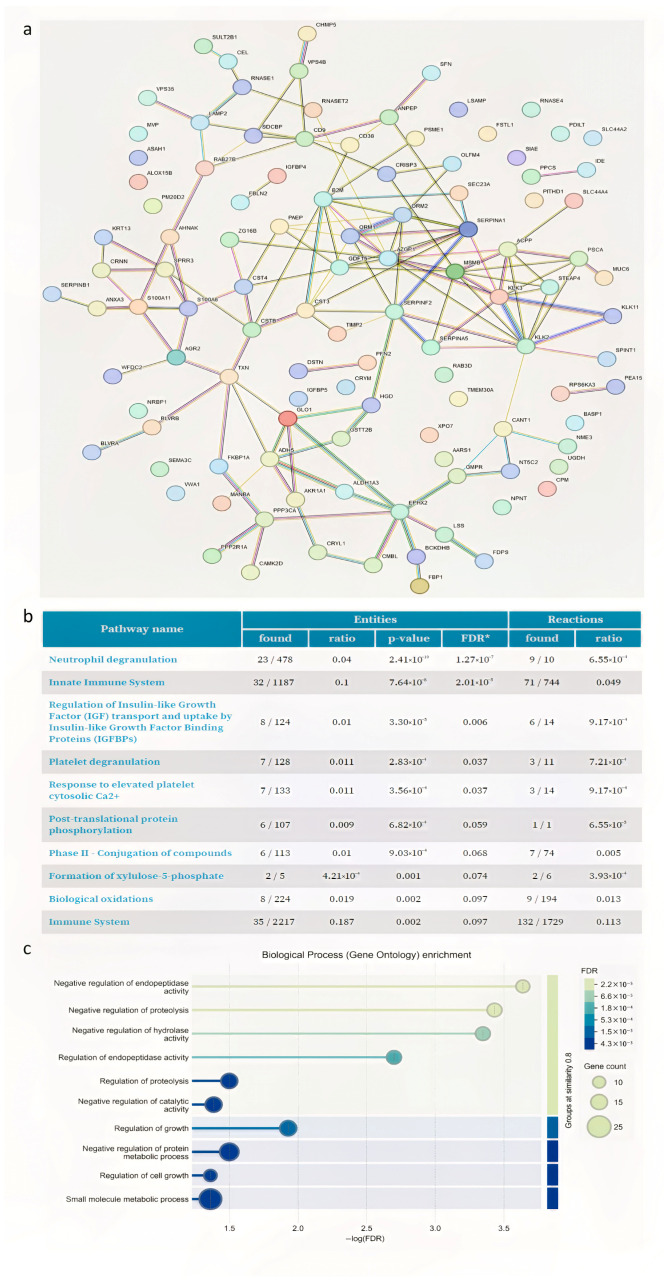
(**a**) STRING analysis shows interaction network of the 115 common proteins upregulated in POST3M and POST6M groups. Each node represents a protein, and each edge represents an interaction. Different colored lines represent seven types of evidence used to predict associations. Red line: fusion evidence; green line: neighborhood evidence; blue line: co-occurrence evidence; purple line: experimental evidence; yellow line: text mining evidence; light blue line: database evidence; and black line: co-expression evidence. (**b**) Table showing the 10 most relevant pathways, sorted by *p*-value, of the 115 common proteins upregulated in POST3M and POST6M groups (pathway enrichment was developed using Reactome database). * False Discovery Rate. (**c**) Graph showing the most relevant cellular component enrichment analysis (gene ontology) of the 115 common proteins upregulated in POST3M and POST6M groups.

**Figure 5 proteomes-13-00034-f005:**
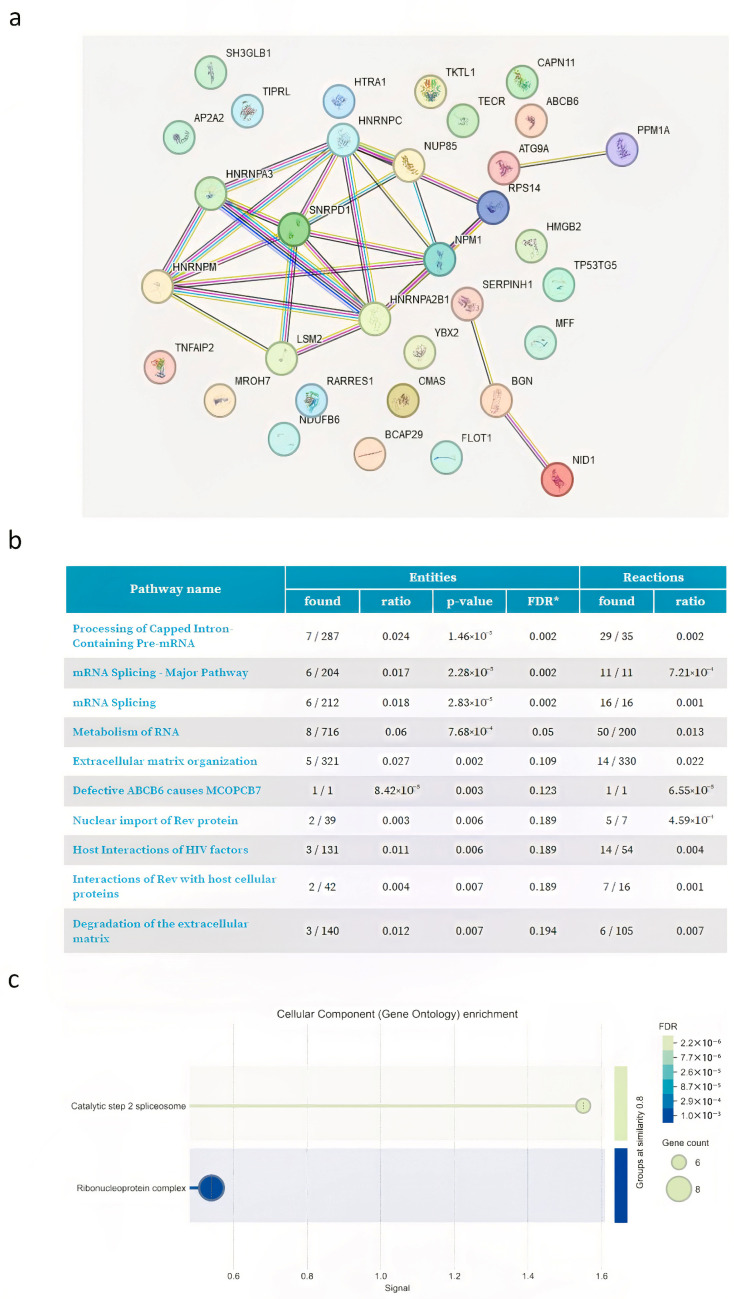
(**a**) STRING analysis shows interaction network of the 33 common proteins downregulated in POST3M and POST6M groups. Each node represents a protein, and each edge represents an interaction. Different colored lines represent seven types of evidence used to predict associations. Red line: fusion evidence; green line: neighborhood evidence; blue line: co-occurrence evidence; purple line: experimental evidence; yellow line: text mining evidence; light blue line: database evidence; and black line: co-expression evidence. (**b**) Table showing the 10 most relevant pathways, sorted by *p*-value, of the 33 common proteins downregulated in POST3M and POST6M groups (pathways enrichment was developed using Reactome database). * False Discovery Rate. (**c**) Graph showing the most relevant cellular component enrichment analysis (gene ontology) of the 33 common proteins downregulated in POST3M and POST6M groups.

**Figure 6 proteomes-13-00034-f006:**
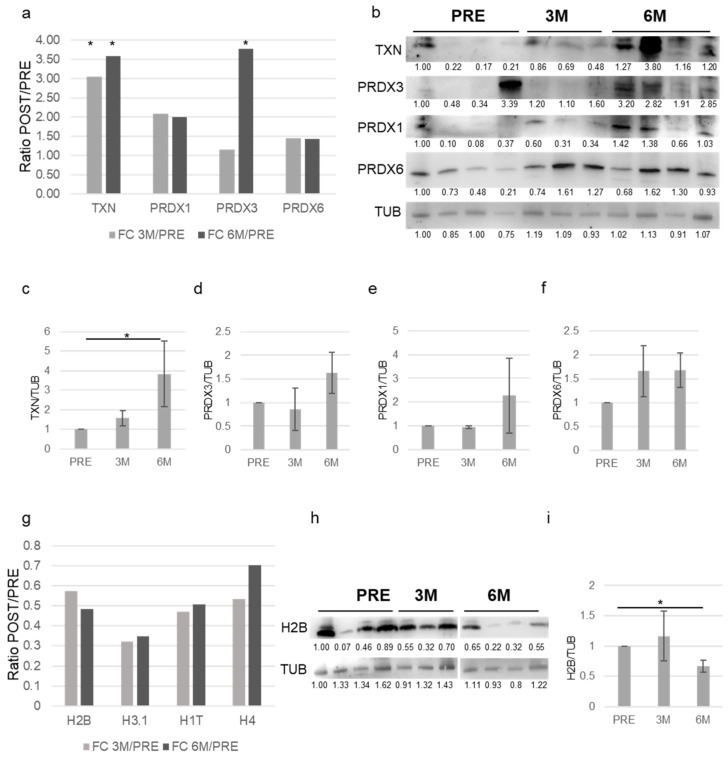
(**a**) Histogram shows the ratio (FC-fold change) of proteomic values of the indicated proteins Pre/3M and Pre/6M. The value of each patient’s pre-sperm sample is arbitrarily set to 1. (* *p* < 0.05). (**b**) Representative Western blot analysis of the indicated proteins in sperm cells from patients Pre, 3 (3M) and 6 months (6M) after surgery (densitometry intensity of each band was shown; the first band was arbitrarily set to 1). (**c**) Histogram shows the ratio of densitometric values of TXN to TUB relative to (**b**). The densitometric value of each patient’s pre-sperm sample is arbitrarily set to 1. (N = 5, * *p* < 0.05, two-tailed unpaired *t*-test). (**d**) Histogram shows the ratio of densitometric values of PRDX3 to TUB relative to B. The densitometric value of each patient’s pre-sperm sample is arbitrarily set to 1. (N = 5). (**e**) Histogram shows the ratio of densitometric values of PRDX1 to TUB relative to (**b**). The densitometric value of each patient’s pre-sperm sample is arbitrarily set to 1. (N = 5). (**f**) Histogram shows the ratio of densitometric values of PRDX6 to TUB relative to (**b**). The densitometric value of each patient’s pre-sperm sample is arbitrarily set to 1. (N = 5). (**g**) Histogram shows the ratio (FC-fold change) of proteomic values of the indicated proteins Pre/3M and Pre/6M. The value of each patient’s pre-sperm sample is arbitrarily set to 1. (**h**) Representative Western blot analysis of the indicated proteins in sperm cells from patients Pre, 3 (3M) and 6 months (6M) after surgery (densitometry intensity of each band was shown; the first band was arbitrarily set to 1). (**i**) Histogram shows the ratio of densitometric values of H2B to TUB relative to (**h**). The densitometric value of each patient’s pre-sperm sample is arbitrarily set to 1. (N = 5, * *p* < 0.05).

**Table 1 proteomes-13-00034-t001:** General characteristics of the population.

Patients	Age	Varicocele Features
A	41	Left, Grade III
B	39	Left, Grade III
C	32	Left, Grade III
D	23	Left, Grade III
E	43	Left, Grade IV

## Data Availability

The datasets used and/or analyzed during the current study are available from the corresponding author upon reasonable request.

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
