# Peer review of "Molecular Remodeling of the Sperm Proteome Following Varicocele Sclero-Embolization: Implications for Semen Quality Improvement"

_proteomes, 2025, doi:10.3390/proteomes13030034_

Round 1

Reviewer 1 Report

Comments and Suggestions for Authors

General Comments

The manuscript presents a timely and relevant investigation into sperm proteomic remodeling following percutaneous varicocele embolization in patients showing improved semen parameters. The integration of clinical data with longitudinal proteomic analysis is a notable strength, and the findings are well aligned with known mechanisms of oxidative stress and male infertility. However, the manuscript requires major revisions before it can be considered for publication. Key concerns include:
- The small sample size (n=5) and use of pooled samples, which limit biological inference;
- Lack of detail in the proteomic data processing pipeline;
- Absence of corrections for multiple comparisons;
- Overinterpretation of some findings as causative rather than associative.

These issues do not undermine the value of the work but do demand greater methodological transparency and interpretive caution.

Specific Comments

Title and Abstract

  • The abstract should mention the number of patients (n=5) and the use of pooled samples.

  • The exploratory nature of the study should be clearly stated.

Introduction

  • The rationale for focusing only on 'good responders' should be better justified.

Materials and Methods

  • Specify the volume of sperm used for each proteomic replicate.

  • Clarify the total number of spermatozoa or the protein concentration input per patient before pooling.

  • Include key parameters for protein identification: search engine, protein database used (e.g., UniProt Homo sapiens), FDR threshold.

  • The strategy for handling missing values is not described.

  • The authors should include in section 2.5 (Statistical and Bioinformatics Analysis) the specific parameters selected for the STRING and Reactome analyses, including the minimum required interaction score used in the protein–protein interaction network construction.

  • State whether the LogFC thresholds used were accompanied by adjusted p-values

Results

  • The distinction between objective observations and biological interpretation should be more explicit.

  • Protein counts across timepoints should include Venn diagram numbers and total IDs.

  • The description provided in the first paragraph of section 3.2 (Lines 268–276) lacks clarity and appears confusing. Rather than functioning as a coherent narrative of the results, it reads more like an extended figure legend. The authors should rephrase this section to clearly describe the main findings in the text, integrating the interpretation of the heatmap, Venn diagram, and DAP histogram into a structured and informative account. Key numerical values and trends should be highlighted and contextualized, rather than relying on figure captions to convey essential information.

  • In the description of Figures 3, 4 and 5, the authors discuss the functional role and biological relevance of several identified proteins directly within the results section. This mixing of data presentation with interpretation and citation of literature is inappropriate for the Results section. These explanatory segments, along with their references, should be relocated to the Discussion section to preserve structural clarity and maintain a strict separation between results and interpretation.

Discussion

  • Overinterpretation: some statements imply causation rather than correlation (e.g., 'varicocelectomy restored chromatin organization').

  • The limitations of the study design (small n, pooling, no control group) are insufficiently acknowledged.

Conclusions

  • Require moderation and clearer framing as exploratory.

  • Recommend including a paragraph on future validation in larger cohorts and non-responder comparisons.

  • The discussion omits a critical analysis of the seminal parameters, which were statistically improved and serve as the clinical foundation for selecting “good responders.” These results deserve explicit discussion, not only to contextualize the biological significance of proteomic changes, but also to explore possible functional links between molecular remodeling and improved sperm motility, concentration, or TMSC. The manuscript would greatly benefit from a more interactive discussion connecting proteomic findings to the observed improvements in semen quality, thereby enhancing both clinical relevance and mechanistic insight.

Figures & Tables

  • Consider adding a graphical abstract or summary figure integrating proteomic and clinical findings.

Abbreviations and Terminology

  • Define abbreviations such as TXN, PRDX, ApoH, TMSC at first mention.

  • Ensure consistent terminology (e.g., varicocelectomy vs. embolization, sample vs. specimen).

Comments on the Quality of English Language

The manuscript is written in generally understandable English; however, several sections would benefit from careful editing to improve clarity, conciseness, and overall readability. In particular, the Results section includes long and ambiguous sentences that obscure the key findings, and the Discussion contains instances of imprecise language that may lead to overinterpretation. A professional language review is recommended to enhance the scientific impact and readability of the manuscript.

Author Response

REV 1

General Comments

The manuscript presents a timely and relevant investigation into sperm proteomic remodeling following percutaneous varicocele embolization in patients showing improved semen parameters. The integration of clinical data with longitudinal proteomic analysis is a notable strength, and the findings are well aligned with known mechanisms of oxidative stress and male infertility. However, the manuscript requires major revisions before it can be considered for publication. Key concerns include:
- The small sample size (n=5) and use of pooled samples, which limit biological inference;
- Lack of detail in the proteomic data processing pipeline;
- Absence of corrections for multiple comparisons;
- Overinterpretation of some findings as causative rather than associative.

These issues do not undermine the value of the work but do demand greater methodological transparency and interpretive caution.

Specific Comments

Title and Abstract

  • The abstract should mention the number of patients (n=5) and the use of pooled samples.

Thanks for your suggestion. We revised the abstract.

  • The exploratory nature of the study should be clearly stated.

Thanks for your suggestion. We revised the abstract.

Introduction

  • The rationale for focusing only on 'good responders' should be better justified.

Thanks for your comment. We revised the introduction section according to your suggestion.

Materials and Methods

  • Specify the volume of sperm used for each proteomic replicate.

Thanks for your comment. We revised the text.

  • Clarify the total number of spermatozoa or the protein concentration input per patient before pooling.

Thanks for your comment. We clarify this information in the Materials and Methods section.

  • Include key parameters for protein identification: search engine, protein database used (e.g., UniProt Homo sapiens), FDR threshold.

Thanks for your comment. LC-MS and MS/MS data were elaborated by Proteome Discoverer software (version 2.4, Thermo Fisher Scientific), based on SEQUEST HT cluster as search engine against the Swiss-Prot Homo Sapiens proteome (UniProtKb, Swiss-Prot, homo+sapiens).The identifications were validated by the Percolator node, with the strict target value of False Discovery Rate (FDR) set at 0.01 and the relaxed value at 0.05. Identification data were filtered for high confidence and peptide rank 1. We add this information in the text.

  • The strategy for handling missing values is not described.

Thanks for your suggestion. To handle missing values, we considered the mean abundance of each protein across the technical triplicates. Only proteins that were quantified in at least two out of the three technical replicates were included in the analysis. This approach was used to ensure data reliability while minimizing the impact of missing values. We add this information in the text.

  • The authors should include in section 2.5 (Statistical and Bioinformatics Analysis) the specific parameters selected for the STRING and Reactome analyses, including the minimum required interaction score used in the protein–protein interaction network construction.

Thanks for your suggestion. We include these information in the section Statistical and Bioinformatics Analysis

  • State whether the LogFC thresholds used were accompanied by adjusted p-values

We appreciate the reviewer’s comment. In our study, we used pooled samples for each experimental condition, and therefore biological replicates were not available. As a result, we were unable to perform statistical testing to calculate adjusted p-values. Consequently, LogFC thresholds were used as descriptive indicators of expression changes rather than for determining statistical significance. We add this information in the text.

Results

  • The distinction between objective observations and biological interpretation should be more explicit.

Thanks for your comment. We revised the text according to your suggestion.

  • Protein counts across timepoints should include Venn diagram numbers and total IDs.

Thanks for your suggestion. The information are included in Supplementary file 1 and was mentioned in the text.

  • The description provided in the first paragraph of section 3.2 (Lines 268–276) lacks clarity and appears confusing. Rather than functioning as a coherent narrative of the results, it reads more like an extended figure legend. The authors should rephrase this section to clearly describe the main findings in the text, integrating the interpretation of the heatmap, Venn diagram, and DAP histogram into a structured and informative account. Key numerical values and trends should be highlighted and contextualized, rather than relying on figure captions to convey essential information.

Thanks for your suggestion. We revised this paragraph to clearly present the main findings, integrating the interpretation of the heatmap, Venn diagram, and DAP histogram into a coherent and informative narrative.

  • In the description of Figures 3, 4 and 5, the authors discuss the functional role and biological relevance of several identified proteins directly within the results section. This mixing of data presentation with interpretation and citation of literature is inappropriate for the Results section. These explanatory segments, along with their references, should be relocated to the Discussion section to preserve structural clarity and maintain a strict separation between results and interpretation.

Thanks for your comment. We revised the text according to your suggestion.

Discussion

  • Overinterpretation: some statements imply causation rather than correlation (e.g., 'varicocelectomy restored chromatin organization').

Thanks for your comment. We revised the discussion according to your suggestion.

  • The limitations of the study design (small n, pooling, no control group) are insufficiently acknowledged.

Thanks for your comment. We revised the text according to your suggestion and we argument the limitations of our study in the Conclusion.

Conclusions

  • Require moderation and clearer framing as exploratory.

Thanks for your comment. We revised the text according to your suggestion.

  • Recommend including a paragraph on future validation in larger cohorts and non-responder comparisons.

Thanks for your comment. We revised the text according to your suggestion and we argument the limitations of our study.

  • The discussion omits a critical analysis of the seminal parameters, which were statistically improved and serve as the clinical foundation for selecting “good responders.” These results deserve explicit discussion, not only to contextualize the biological significance of proteomic changes, but also to explore possible functional links between molecular remodeling and improved sperm motility, concentration, or TMSC. The manuscript would greatly benefit from a more interactive discussion connecting proteomic findings to the observed improvements in semen quality, thereby enhancing both clinical relevance and mechanistic insight.

Thanks for your suggestion. We argument this point in the Discussion section.  

Figures & Tables

  • Consider adding a graphical abstract or summary figure integrating proteomic and clinical findings.

Thanks to your suggestion. We add a graphical abstract to the manuscript.

Abbreviations and Terminology

  • Define abbreviations such as TXN, PRDX, ApoH, TMSC at first mention.

Thanks for your comment. We revised the text.

  • Ensure consistent terminology (e.g., varicocelectomy vs. embolization, sample vs. specimen).

Thanks for your comment. We revised the text.

Comments on the Quality of English Language

The manuscript is written in generally understandable English; however, several sections would benefit from careful editing to improve clarity, conciseness, and overall readability. In particular, the Results section includes long and ambiguous sentences that obscure the key findings, and the Discussion contains instances of imprecise language that may lead to overinterpretation. A professional language review is recommended to enhance the scientific impact and readability of the manuscript.

Reviewer 2 Report

Comments and Suggestions for Authors

This manuscript explores the proteomic remodeling in sperm samples of varicocele patients before and after percutaneous scleroembolization, highlighting molecular mechanisms linked to improved semen quality. Through a combination of semen analysis, LC-MS/MS-based label-free proteomics, STRING network analysis, and Western blot validation, the authors identify differentially abundant proteins (DAPs) with functional relevance to sperm function, oxidative stress, and testicular homeostasis.

The topic is timely and relevant, contributing valuable insights into the molecular consequences of varicocele treatment and its positive outcomes on male fertility. The experimental design, proteomics workflow, and validation strategies are solid. However, a few clarifications and improvements are required before final acceptance.

Major Comments

  1. Sample Size and Generalizability

    • The study includes only five patients, all characterized as “responders.” This limitation, although acknowledged by the authors, strongly restricts generalizability. The authors should clarify:

      • Were any “non-responders” excluded?

      • Can they comment further on inter-individual variability and how pooling was justified?

  2. Label-Free Quantification (LFQ) Strategy

    • The authors employed a well-established LFQ method using DDA-based Orbitrap mass spectrometry. While the technical setup is adequate, please provide more information on:

      • Whether normalization across samples was based on total ion current or another strategy.

      • How missing values were handled.

      • Whether any quality control metrics (e.g., CVs among triplicates) were assessed.

  3. Proteomics Data Interpretation

    • While the functional enrichment and STRING-based network analyses are informative, some of the conclusions are too speculative (e.g., links to fertilization or chromatin remodeling). Please moderate the tone or include appropriate references to support functional inferences (e.g., “EPHX2 affects acrosomal integrity…”).

    • Figures 3–5 (STRING plots and pathway tables) could be strengthened with more quantitative annotation, such as FDR values or enrichment scores.

  4. Western Blot Validation

    • The authors successfully validated several key DAPs (e.g., TXN, PRDX1, H2B), adding credibility to their proteomics data.

    • Please clarify:

      • How protein load normalization was performed (e.g., with TUB or total protein stain)?

      • Why some Western blot validations were shown for N = 3 rather than N = 5?

  5. Functional Clustering and Interpretation

    • The authors identified upregulated proteins related to antioxidant defense (e.g., PRDX3, GPX3, TXN) and downregulated proteins linked to mRNA processing and ribonucleoproteins (e.g., SRSF2, NCBP1). The contrasting biological themes are interesting but need deeper discussion—what might downregulation of nuclear RNA binding proteins indicate about sperm maturation?

Minor Comments

  • Language and Style:
    The manuscript is well-written overall, but some sentences are overly long or redundant. A careful language edit is recommended for conciseness and clarity (e.g., repeated mention of TXN/PRDX without functional differentiation).

  • Figures:

    • Improve resolution for STRING diagrams (Figures 3–5) to enhance readability.

    • Some bar graphs lack proper y-axis labels or units.

    • Consider adding summary tables for top 10 up- and down-regulated proteins with fold change and adjusted p-values.

  • Statistical Details:

    • Please specify how p-values were corrected for multiple testing in the DAP analysis (e.g., FDR, Benjamini-Hochberg).

    • Add details on whether significance thresholds (e.g., log2FC > 1.5) were chosen empirically or based on statistical criteria.

  • References:

    • Ensure recent relevant proteomics studies on male infertility are cited, particularly those using label-free quantitation or sperm proteome dynamics.

1. Figure1e, the intercept on the column of PzA did not show on the y-axis, have to revise it to show the real number;

2. Figure2a, there are two arrow icons besides the scale bar. Also, why only N=3 for pre, post 3m, post 6m separtely. In the main text, there are 5 patients you showed. 

3. Figure 3,4,5, the resolution is low, it looks that is the screenshot from the webpage of STRING. Please download the original version with high resolution.

4. Figure 6b,  n=4 in PRE, n=3 in post 3M, n=4 in post 6M, however, several bands on Tublin as the control, did not show the comparable level, especially, tthe fourth band in PRE, the third band in POST 6M. They should be excluded. 

5. Figure 6h,  the band of H2B on 2nd rep in PRE condition, show the totally different amount with the other 3 rep in PRE. Also, the 4 replicates on POST 6M, the huge variablity of 4 replicates. The repeated experiment is required. 

Author Response

REV 2

Comments and Suggestions for Authors

This manuscript explores the proteomic remodeling in sperm samples of varicocele patients before and after percutaneous scleroembolization, highlighting molecular mechanisms linked to improved semen quality. Through a combination of semen analysis, LC-MS/MS-based label-free proteomics, STRING network analysis, and Western blot validation, the authors identify differentially abundant proteins (DAPs) with functional relevance to sperm function, oxidative stress, and testicular homeostasis.

The topic is timely and relevant, contributing valuable insights into the molecular consequences of varicocele treatment and its positive outcomes on male fertility. The experimental design, proteomics workflow, and validation strategies are solid. However, a few clarifications and improvements are required before final acceptance.

Major Comments

  1. Sample Size and Generalizability
  • The study includes only five patients, all characterized as “responders.” This limitation, although acknowledged by the authors, strongly restricts generalizability. The authors should clarify:
    • Were any “non-responders” excluded?

Thanks for your comment. We revised the introduction section to better clarify our aims. Furthermore, the molecular signature associated with successful varicocele treatment will be validated in a future study involving a larger patient cohort and compared with the proteomic profiles of non-responders.

  • Can they comment further on inter-individual variability and how pooling was justified?

Thanks for your comment. Pooling samples in proteomic analysis can be a useful strategy to reduce inter-individual variability and increase statistical power, especially when dealing with a limited number of samples or high variability between individuals. However, it's crucial to be aware that pooling can also lead to the loss of some information, particularly regarding low-abundance proteins or individual variations. We add this comment in the text.

  1. Label-Free Quantification (LFQ) Strategy
  • The authors employed a well-established LFQ method using DDA-based Orbitrap mass spectrometry. While the technical setup is adequate, please provide more information on:
    • Whether normalization across samples was based on total ion current or another strategy.

Thanks for your comment. In our analysis, normalization across samples was performed using the area of unique peptide peaks, which were normalized to account for variability between samples. This strategy allows for more accurate quantification by focusing on peptides uniquely assigned to each protein. We add this information in the text.

  • How missing values were handled.

Thanks for your comment. To handle missing values, we considered the mean abundance of each protein across the technical triplicates. Only proteins that were quantified in at least two out of the three technical replicates were included in the analysis. This approach was used to ensure data reliability while minimizing the impact of missing values. We add this information in the text.

  • Whether any quality control metrics (e.g., CVs among triplicates) were assessed.

Thanks for your comment. Please look at the previous response.

  1. Proteomics Data Interpretation
  • While the functional enrichment and STRING-based network analyses are informative, some of the conclusions are too speculative (e.g., links to fertilization or chromatin remodeling). Please moderate the tone or include appropriate references to support functional inferences (e.g., “EPHX2 affects acrosomal integrity…”).

Thanks for your comment. We revised the manuscript according to your suggestion.

  • Figures 3–5 (STRING plots and pathway tables) could be strengthened with more quantitative annotation, such as FDR values or enrichment scores.

Thanks for your comment. We added this information in Material and Method’s section.

  1. Western Blot Validation

The authors successfully validated several key DAPs (e.g., TXN, PRDX1, H2B), adding credibility to their proteomics data.

Please clarify:

  • How protein load normalization was performed (e.g., with TUB or total protein stain)?

Thanks for your comment. We used Tubulin signal as loading control; our data are showed as ratio between specific protein signal and tubulin.

  • Why some Western blot validations were shown for N = 3 rather than N = 5?

Thanks for your comment. We apologize for the mistake, we analyse all five patients proteins signals.We correct this information in the figure legend.

  1. Functional Clustering and Interpretation
  • The authors identified upregulated proteins related to antioxidant defense (e.g., PRDX3, GPX3, TXN) and downregulated proteins linked to mRNA processing and ribonucleoproteins (e.g., SRSF2, NCBP1). The contrasting biological themes are interesting but need deeper discussion—what might downregulation of nuclear RNA binding proteins indicate about sperm maturation?

Thanks for your comment. We revised the manuscript according to your suggestion in the Discussion section.

Minor Comments

  • Language and Style:
    The manuscript is well-written overall, but some sentences are overly long or redundant. A careful language edit is recommended for conciseness and clarity (e.g., repeated mention of TXN/PRDX without functional differentiation).

Thanks for your comment. We revised the manuscript.

  • Figures:
    • Improve resolution for STRING diagrams (Figures 3–5) to enhance readability.

Thanks for your comment. We improve the resolution of the images according to your suggestion.

  • Some bar graphs lack proper y-axis labels or units.

Thanks for your comment. We revised the figures.

  • Consider adding summary tables for top 10 up- and down-regulated proteins with fold change and adjusted p-values.

The complete lists of differentially abundant proteins (DAPs) with fold changes at 3 and 6 months after treatment are provided in Supplementary Files 3 and 4. In the manuscript, we focus our analysis on the proteins that were commonly up- or downregulated at both time points, as we consider this subset to be of particular clinical relevance.

Statistical Details:

  • Please specify how p-values were corrected for multiple testing in the DAP analysis (e.g., FDR, Benjamini-Hochberg).

Thanks for your comment. In the Differentially Abundant Proteins (DAP) analysis, p-values were corrected for multiple testing using the Benjamini-Hochberg procedure to control the False Discovery Rate (FDR). We add this information in the text.

  • Add details on whether significance thresholds (e.g., log2FC > 1.5) were chosen empirically or based on statistical criteria.

Thanks for your comment. The significance threshold of log2FC > 1.5 was chosen empirically, based on commonly used cut offs in the literature for identifying biologically meaningful changes in protein expression. We add this information in the text.

  • References:
    • Ensure recent relevant proteomics studies on male infertility are cited, particularly those using label-free quantitation or sperm proteome dynamics.

Thanks for your suggestion. We reviewed the literature and added a recent reference to the text.

Additional comments of reviewer 2:

  1. Figure1e, the intercept on the column of PzA did not show on the
    y-axis, have to revise it to show the real number;

Thanks for your comment. We revised the figure.

  1. Figure2a, there are two arrow icons besides the scale bar. Also, why
    only N=3 for pre, post 3m, post 6m separtely. In the main text, there
    are 5 patients you showed.

Thanks for your comment. We revised the figure and delete the arrows. Moreover, the heatmap displays protein levels for each treatment group (Pre, 3M, and 6M), based on pooled samples from five patients per group, each analyzed in analytical triplicate.

  1. Figure 3,4,5, the resolution is low, it looks that is the screenshot
    from the webpage of STRING. Please download the original version with
    high resolution.

Thanks for your comment. We replaced them.

4. Figure 6b,  n=4 in PRE, n=3 in post 3M, n=4 in post 6M, however,
several bands on Tublin as the control, did not show the comparable
level, especially, the fourth band in PRE, the third band in POST 6M.
They should be excluded.

Thanks for your comment. Figure 6b shows a representative Western blot (WB) of selected sperm samples from patients before and after treatment. Tubulin was used as a loading control, and the observed variability in band intensity reflects intrinsic inter-individual differences. Therefore, the quantification of TXN, PRDXs, and H2B was performed by calculating the ratio of each protein to tubulin.

  1. Figure 6h, the band of H2B on 2nd rep in PRE condition, show the
    totally different amount with the other 3 rep in PRE. Also, the 4
    replicates on POST 6M, the huge variablity of 4 replicates. The repeated
    experiment is required.

 Thanks for your comment. We replace this part of WB with repeated experiment.

Round 2

Reviewer 1 Report

Comments and Suggestions for Authors

The revised manuscript presents a clearer and more structured account of the proteomic changes following varicocele embolization in good responders. The authors appropriately added methodological details, such as protein input volumes, database specifications, FDR thresholds, and the handling of missing values, which improve the transparency and reproducibility of the study. The distinction between results and interpretation is now more evident, and the discussion more effectively connects the proteomic findings with clinical outcomes.

The inclusion of a graphical abstract and the expanded discussion on semen quality parameters and study limitations are commendable. However, a professional English language revision is still recommended to address persistent issues with overly long and complex sentences, particularly in the Discussion section. Improving language clarity would enhance the overall readability and scientific impact of the work.

Comments on the Quality of English Language

The manuscript is written in generally understandable English; however, several sections would benefit from careful language editing to improve clarity, reduce redundancy, and enhance scientific readability. In particular, the Results and Discussion sections include long and complex sentences that obscure the main findings or introduce interpretative ambiguity. A professional English language review is recommended to ensure greater fluency, conciseness, and precision of expression.